# Determinants of Actual COVID-19 Vaccine Uptake in a Cohort of Essential Workers: An Area-Based Longitudinal Study in the Province of Prato, Italy

**DOI:** 10.3390/ijerph192013216

**Published:** 2022-10-14

**Authors:** Vieri Lastrucci, Chiara Lorini, Lorenzo Stacchini, Enrica Stancanelli, Andrea Guida, Alessio Radi, Chiara Morittu, Salvatore Zimmitti, Giorgia Alderotti, Marco Del Riccio, Angela Bechini, Sara Boccalini, Guglielmo Bonaccorsi

**Affiliations:** 1Epidemiology Unit, Meyer Children’s Hospital, 50139 Florence, Italy; 2Department of Health Sciences, University of Florence, Viale GB Morgagni 48, 50134 Florence, Italy; 3Medical Specialization School of Hygiene and Preventive Medicine, University of Florence, 50134 Florence, Italy

**Keywords:** COVID-19, vaccine, vaccination uptake, predictors, health literacy, infection prevention behaviors, risk perception, knowledge, attitudes and practices, area-based

## Abstract

Identifying determinants of COVID-19 vaccine uptake is essential for developing effective strategies for promoting vaccination. This longitudinal study aimed to explore predictors of actual COVID-19 vaccine uptake in workers involved in essential services during the first lockdown period in the Prato Province (Italy). All essential workers were invited and surveyed before COVID-19 vaccine approval (96.5% participation rate). Participants were followed up to evaluate their actual COVID-19 vaccination uptake using the vaccination register. Multinomial models were performed to assess predictors of delayed vaccination or non-vaccination. A total of 691 participants were included, of whom 21.7% had delayed the vaccination and 4.4% were unvaccinated. Participants with a sufficient level of health literacy were 50.2% in the vaccinated-on-time group and 32.3% in the unvaccinated group. The multinomial model predictors of delayed vaccination were work type (OR = 0.51), age between 50 and 59 years (OR = 1.82), and influenza vaccination uptake in the last season (OR = 2.51). Predictors of being unvaccinated were work type (OR = 0.33) and attitudes related to attributing less importance to COVID-19 preventive measures (OR = 0.47). Findings showed distinct predictors for COVID-19 vaccination delay and for being unvaccinated. Being unvaccinated seems to be associated with a general skepticism toward prevention measures.

## 1. Introduction

Over the past century, vaccinations have become a routine and effective preventive measure against viral diseases [1]. It has been widely demonstrated that vaccination provides immunity and prevents diseases among vaccinated individuals; moreover, vaccines have been reducing infections even among individuals who are not vaccinated, by creating community protection [2]. Nevertheless, during the COVID-19 vaccination campaign, the phenomenon of vaccine hesitancy generated a delay in acceptance or refusal of vaccines [3,4,5].

Italy has been one of the first Western countries to be severely hit by the COVID-19 pandemic [6,7]. On 20 February 2020, the first local Italian case was confirmed [8,9]. To contain the spread of the virus, a national lockdown (10th March–4th May 2020) was established with the closure of all non-essential activities. The first approved vaccines became available on 27th December 2020 [10]. A vaccination campaign was organized according to the “National Strategic Vaccine Plan for the Prevention of SARS-CoV-2 Infections” [11] which—due to the limited initial availability of vaccine doses—gave priority to the elderly and the frail. As more vaccines and doses became available, the vaccination campaign gradually opened the possibility of being vaccinated to younger age groups until the opening for people older than 16 years in June 2021.

The target of a high COVID-19 vaccination coverage rate was threatened by vaccine hesitancy. The WHO defined vaccine hesitancy as a “delay in acceptance or refusal of vaccination despite the availability of vaccination services”, influenced by several factors including issues of confidence (do not trust vaccine or provider), complacency (that is people who do not perceive a need for a vaccine, do not value the vaccine), and convenience (in terms of easiness of access to vaccination). Vaccine-hesitant individuals are a heterogeneous group who show varying degrees of indecision about specific vaccines or vaccination in general [12,13].

Various factors have been associated with COVID-19 vaccine acceptance, hesitancy and refusal [14,15,16,17,18]. Yet, most of the COVID-19 literature mainly focused on vaccine uptake intention or self-reported vaccine uptake, and it has been shown that the willingness to receive a COVID-19 vaccine may not predict later vaccine uptake [19]. A recent review showed how gender, educational level, influenza vaccination history, and trust in the government were fundamental predictors of COVID-19 vaccination acceptance, whereas concerns about side effects, mistrust in the government, and religious beliefs were key factors in predicting vaccine hesitancy [16]. Yet, an association with a specific age category has not been proven. Another predictor can be the employment status: some studies found that some occupations, for example, healthcare workers, are less hesitant to COVID-19 vaccination [20]. However, other studies showed that employment status was not significantly associated with vaccination willingness [21]. Health literacy (HL) may help to detect fake news and misbeliefs about vaccine efficacy and safety; however, to date, evidence on the relationship between HL and vaccination uptake is inconsistent [22,23]. As for the specific case of HL and COVID-19 vaccination, to date only limited research has been specifically performed, and this research had considered only the intention to get vaccinated and not the actual vaccination uptake [24]. Furthermore, it is still unclear whether vaccine hesitancy and vaccine refusal correlate with the knowledge, attitudes, and adherence to other COVID-19 prevention practices and to COVID-19 risk perception.

This longitudinal study aims to evaluate determinants of actual COVID-19 vaccination uptake in an area-based cohort of workers involved in essential services during the first lockdown period in the Prato Province (Italy). As vaccination delay and refusal may be linked to different predictors, vaccination uptake was examined considering these outcomes separately. This study examined several potential predictors at different levels: i. socio-demographic factors; ii. risk factors for COVID-19; iii. health literacy; iv. knowledge, attitudes, and practices related to COVID-19 prevention measures; and v. COVID-19 risk perception.

## 2. Materials and Methods

The study was approved by the Local Ethics Committee and was conducted according to the Helsinki Declaration

### 2.1. Study Design and Population

This is a prospective cohort study conducted on a population-based sample in the Prato Province (Tuscany Region, Italy). The study sample included people aged 18 years or older involved in different essential activities during the first lockdown period (11 March to 3 May 2020) in Italy. All the workers and volunteers of the Civil Protection and public employees of Prato Province and of the municipalities of Cantagallo, Carmignano, Montemurlo, Poggio a Caiano, Vaiano, and Vernio were invited to participate in the study. All the workers working from home during the first lockdown period were excluded from the study. Further details on the enrolled workers and volunteers as well as of the study design of the cross-sectional phase have been described elsewhere [23,25,26].

### 2.2. Data Collection

Participants were asked to respond to a structured interview administered by the research team. Interviews were conducted from May to June 2020. The 67-item survey lasted about 25 min. The survey explored the following areas: i. sociodemographic characteristics; ii. HL level; iii. risk conditions for severe COVID-19; iv. knowledge, attitudes, and practices (KAP) towards COVID-19 prevention measures and risk perception for COVID-19; v. Influenza vaccination uptake in the 2019–2020 season.

Participants were followed up in order to evaluate their COVID-19 vaccination uptake. Data related to vaccination and SARS-CoV-2 infection were collected from the Collective Prevention Health Information System (CPHIS), a regional system that collects data from all the COVID-19 vaccinations and infections occurring in Tuscany. The follow-up started from the date of participant enrollment and ended on the 29 March 2022. Data on all the vaccination doses administered to the study participants were retrieved; in particular, for each vaccine dose, the date of administration and the type of vaccine administered were collected. As for COVID-19 infection, CPHIS collected results from all the nasal swabs performed in Tuscany; for each participant who had had COVID-19 during the follow-up period, the date of the first positive nasal swab was collected.

### 2.3. Outcome Definition

The primary outcome of the study was to assess COVID-19 vaccination uptake by categorizing the study population into the following groups: vaccinated on time, delayed vaccination, and unvaccinated. A delayed vaccination was defined as a vaccine first dose which occurred at least 21 days or more from the date of reservation opening. The date of the vaccine reservation opening for each age group was taken into account in defining the vaccination timeliness of the study participants; Table 1 reports the date of reservation opening according to population age in the Tuscany region. Furthermore, according to the Italian regulations, people who had contracted COVID-19 were asked to postpone vaccination for 6 months from the first positive nasal swab; to take into account this regulation, this group was considered having delayed the COVID-19 vaccination in case the first shot occurred after six months plus 21 days from the date of the first positive nasal swab. Participants who had not been vaccinated against COVID-19 at the end of the follow-up period were considered as unvaccinated.

The following independent variables were considered: age, sex, education level, nationality, influenza vaccination uptake in the 2019–2020 season, household conditions (living with people older than 64 years or with people with chronic health conditions, number of cohabitants), HL level, smoking habit, risk conditions for severe COVID-19, KAP towards COVID-19 prevention measures, and risk perception for COVID-19. The following risk conditions were considered: diabetes, overweight and obesity, heart diseases, pulmonary diseases, diseases of the immune system, chronic kidney diseases, chronic liver diseases, organ or bone marrow transplantation, chronic neurological diseases, oncological diseases in the last 5 years, hematological diseases, and major surgery intervention that required general anesthesia during the last year. Risk conditions were dichotomized in “no risk condition” and “one or more risk conditions”. As for HL, the short form of the European Health Literacy Survey Questionnaire HLS-EU-Q6 validated for the Italian language was used [24].

As for the KAP towards COVID-19 prevention measures and risk perception for COVID-19, the items in the questionnaire were developed with a pre-testing methodology and are described in a previous work [26]. Self-reported level of knowledge on COVID-19 preventive behaviors was investigated (possible responses: “very poor”, “poor”, “fair”, “good”, and “very good”; score from 1 to 5). As for attitudes, the importance attributed to several COVID-19 preventive measures was asked (possible responses: “low importance”, “slightly important”, “neutral”, “moderately important”, and “very important”; score from 1 to 5). As for the practices, the compliance with the basic recommendations for preventing COVID-19 was investigated (response options: “never”, “almost never”, “occasionally/sometimes”, “almost every time”, and “every time”; score 1 to 5). Lastly, risk perception for COVID-19 was evaluated in terms of disease susceptibility (self-rated likeliness of contracting COVID-19) and fear of COVID-19 severe health consequences for oneself and for family members. Response options were: “not at all”, “slightly”, “moderately”, “very”, and “extremely”; score 1 to 5.

### 2.4. Statistical Analyses

Continuous variables were presented as mean (standard deviation—SD) and median (interquartile range—IQR). Categorical variables were presented as frequency and percentage. The normality of numerical variables was tested using the Shapiro–Wilk test. The association between the outcome and the collected variables was tested with Fisher’s exact test, Chi-square test, and the Kruskal–Wallis test, as appropriate.

For the items related to the attitudes toward COVID-19 prevention measures and risk perception, two overall scores were calculated; i.e., overall attitude score and overall risk perception. The overall attitude and risk perception scores were calculated as the mean of the scores of the attitude and risk perception items, respectively. HL was analyzed as a continuous variable (i.e., HL-EU-Q6 score) and categorical variable using the level of HL (inadequate, problematic, and sufficient HL) as defined in the validation study [27].

A multivariate multinomial logistic regression model was performed to assess predictors of COVID-19 vaccination uptake (vaccinated on time, delayed vaccination, and unvaccinated), using the vaccinated-on-time group as the reference. All the variables that were significantly associated with the outcome at the univariate analyses were included in the model. 

All the analyses were performed using R 4.2.1 (RStudio: Integrated Development for R. RStudio, Inc., Boston, MA, USA), considering an alpha level of 0.05 as significant.

## 3. Results

A total of 751 people filled out the questionnaire, with a study participation rate of 96.5%. Record linkage with vaccine and COVID-19 registries was not possible for 60 participants. The characteristics refer to a total of 691 participants, shown in Table 2.

Most participants were vaccinated on time, although civil protection workers were vaccinated on time more frequently than public employees (79.3% vs 61.2%), moreover, civil protection workers were less frequently unvaccinated (3.3%) than public employees (7.2%) (*p* < 0.001). The median age of the total sample was 51 years (IQR: 37–61; range 18–84), age was significantly different among the three vaccination subgroups (*p* = 0.02): 51 years in the vaccinated-on-time group (IQR: 41–58; range: 18–84), 52 years in the delayed vaccination group (IQR: 36–62; range: 18–79), and 47 years in the unvaccinated group (IQR: 38–65.5; range: 18–80). In particular, the highest proportion of unvaccinated people was registered in the 40–49 years age group, the highest proportion with a delayed vaccination was registered in the 40–49 and 50–59 years age groups, while participants aged 18–39 years and those aged 60 years or higher showed the highest proportion of on-time vaccination.

The majority of the subjects were Italian, with a high education level, never smokers, and had not received a previous flu vaccination. Participants who had a flu vaccination in the 2019/2020 season were more frequently vaccinated on time and less frequently unvaccinated for COVID-19 (*p* < 0.001).

Regarding HL level, 48.8% of the population had a sufficient HL, 30.8% had a problematic level, and 6.7% had an inadequate level (Table 3).

Participants vaccinated on time had a sufficient HL level more frequently compared with those in the delayed vaccination group and with those unvaccinated. Furthermore, for a relevant proportion of unvaccinated participants, it was not possible to calculate the HL level for missing data in one or more items. As far as the HL score is concerned, the mean HL score showed a decreasing trend passing from the vaccinated-on-time, to the delayed vaccination, to the unvaccinated group, although this trend was not significant.

Table 4 reports the descriptive analysis of the items related to KAP towards COVID-19 preventive measures and COVID-19 risk perception. Although a decreasing trend in the level of self-reported knowledge of prevention measures to protect from and avoid transmitting COVID-19 was recorded when passing from participants vaccinated on-time to those who delayed vaccination or were unvaccinated, no significant difference was observed among the three subgroups. Concerning the attitudes towards COVID-19 preventive measures, there was a significant difference among the three subgroups in the following items: wearing a face mask, staying at home as much as possible, avoiding meeting friends/family members from another household, except for reasons of necessity, and in the overall attitude score. In particular, participants vaccinated on time considered these prevention measures more relevant than the other two subgroups (e.g., overall attitude score: vaccinated on time, mean: 4.67 ± 0.5 median: 5 (4.4–5); delayed vaccination, mean: 4.65 ± 0.5, median: 5 (4.4–5); unvaccinated, mean: 4.35 ± 0.7, median: 4.6 (3.9–5). Similarly, participants vaccinated on time reported the highest adherence to prevention practices towards COVID-19 (vaccinated on time, mean: 4.56 ± 0.63, median: 5 (4–5); delayed vaccination, mean: 4.46 ± 0.66, median: 5 (4–5); unvaccinated, mean: 4.34 ± 0.56, median: 4.6 (3.9–5), with a significant difference among the three groups. Lasty, there was no significant difference in term of COVID-19 risk perception among the three groups.

Table 5 reports the results of the multinomial regression model evaluating the predictors of delayed vaccination and of being unvaccinated.

Compared with participants working as public employees, working as public volunteers of the civil protection emerged as a protective factor for a delayed vaccination (OR 0.51; *p* = 0.002) and for being unvaccinated (OR 0.33; *p* = 0.009).

Risk factors most strongly associated with COVID-19 delayed vaccination were: belonging to the 50–59 year-old class (OR 1.82; *p* = 0.03) and being unvaccinated for influenza in the 2019–2020 season (OR 2.51; *p* = 0.002). 

Lastly, having a higher overall attitude score emerged as a protective factor for being unvaccinated (OR 0.47; *p* = 0.01).

## 4. Discussion

This study aimed to explore predictors of SARS-CoV-2 vaccine uptake in a representative sample of workers of the Civil Protection and public employees involved in different essential activities during the first lockdown period in Italy. COVID-19 vaccination uptake was significantly associated with several attitudes related to COVID-19 preventive measures; in particular, the rate of importance attributed to COVID-19 preventive measures decreased passing from participants vaccinated on time, to those who delayed vaccination and to those unvaccinated. Instead, COVID-19 risk perception was not significantly associated with vaccination uptake. As for HL, although not significantly associated with vaccination uptake, a decreasing trend in the proportion of participants with a sufficient level of HL was found passing from the vaccinated-on-time group, to the delayed vaccination, to the unvaccinated. The results of the multinomial regression model showed that predictors correlated with a delayed vaccination were: worker subgroups, age (50–59 years), and influenza vaccination uptake in the last flu epidemic season; while predictors correlated with being unvaccinated were worker subgroups and attitudes towards COVID-19 preventive measures.

Our findings showed a relatively high vaccine uptake among essential workers, especially when compared to the data from the Tuscany region’s general population and other population subgroups, such as unemployed people [28,29,30]. In particular, participants involved in the civil protection services showed the highest rate of vaccine uptake. This may be explained as essential workers faced the early stages of the pandemic and their ability to understand the importance of vaccination may have been enhanced by events experienced during such a dramatic period, while other population subgroups may have lacked such kind of experiences.

So far, few studies have evaluated the role of HL in predicting vaccination uptake [22], especially in the context of COVID-19 vaccination [24]. The intention to get vaccinated against COVID-19 has been reported to be associated with the ability to detect fake news and HL [24], but this outcome may be biased, for example, by social desirability; therefore, evidence from studies considering the actual vaccination uptake is needed. When considering other vaccines, the correlation between HL and vaccination uptake has been shown to be inconsistent (sometimes positive, sometimes negative, or absent) [22]. This inconsistent association is probably linked to the fact that vaccination uptake is the result of a complex decision-making process that general health literacy level cannot fully explain; for this reason, the concept of “Vaccine Literacy” (VL) [31,32] has been proposed on the same idea of Health Literacy: “it is not simply knowledge about vaccines, but it entails motivation and competence to deal with information about immunization, diseases prevention and also health promotion” [33].

Study findings showed an interesting and significant trend concerning the attitude toward COVID-19 prevention measures. In particular, the unvaccinated group showed the most negative feelings regarding prevention measures, while the vaccinated-on-time group had the most positive feelings. Unvaccinated people, therefore, seem to show a broader behavior characterized not only by vaccine refusal but also by a general higher level of disregard and for basic social and prevention norms [14].

Interestingly, in our study, knowledge of prevention measures and risk perception of COVID-19 were not predictors of vaccination uptake. Although a preliminary study of COVID-19 vaccine acceptability conducted in the US [34] before vaccine development showed that people with higher knowledge and risk perception of COVID-19 were more likely to be willing to get vaccinated, it may be hypothesized that—after the approval of COVID-19 vaccines—other factors related to concerns about vaccine safety and/or to a diminished perception of the severity of the disease during the subsequent pandemic waves may have influenced the actual vaccine uptake.

Higher adherence to COVID-19 vaccination and a timelier vaccination were observed in workers and volunteers of the Civil Protection. This is probably related to higher attention to health risks and a stronger collective responsibility linked with their activities in the field of healthcare.

In line with our study findings, previous influenza vaccination uptake has been suggested as a strong predictor of COVID-19 vaccine acceptance [35,36]. Furthermore, in the Italian population, propensity toward influenza vaccination has been shown to be a significant predictor of COVID-19 vaccination uptake [37]. This finding probably reflects a more generalized propensity to vaccine acceptance linked to a higher level of trust in the effectiveness and safety of vaccines in this population subgroup [38].

In our study, the age group that was mostly at risk of delayed vaccination was the 50–59 year-old class, which is in accordance with Tuscany region data, where the COVID-19 vaccine coverage by age group showed a U-shaped curve with the lowest level in the 40–59 years age group [39]. This finding is in line with a recent study [40] which showed—among healthcare students and professionals—that “Generation X”—i.e., people aged between 42 and 57 years old—had the most negative attitudes towards vaccination.

The present study had several strengths and limitations. This is a population-based study with a high participation rate, so it can be considered representative of people who made that kind of activity in all the study area. The follow-up was based on highly reliable data, which were collected from the CPHIS of the Tuscany region; this is the official register collecting data from all the COVID-19 vaccinations and infections that occurred in the region. As far as the limitations are concerned, it should be highlighted that questions regarding KAPs and risk perception were collected only at the beginning of the study, during the very first phase of the pandemic. Therefore, they may have changed during the subsequent pandemic waves, due to the emergence of new viral variants or owing to the effect of pandemic fatigue [41]. KAP items were reported by the participants, and thus the responses may have been affected by a social desirability bias. However, participants were informed that their answers would be managed and analyzed anonymously. Moreover, it is important to note that, although the attitude and risk perception scores were based on items frequently used in the literature, they were not previously validated. Another limitation is related to the external validity of the study findings as the study sample included only essential workers, so our findings are generalizable only to this population subgroup. Lastly, CPHIS is a regional register, data on vaccination or infections occurring in other Italian regions or abroad may have not been registered if the participants did not transmit the vaccination certificate, or their condition of having been sick with COVID-19, to the regional authorities.

## 5. Conclusions

The present study investigated predictors of delayed COVID-19 vaccination and of being unvaccinated in a representative sample of essential workers. Study findings showed distinct predictors for vaccination delay and for being unvaccinated; in particular, being unvaccinated seems to be associated with a more general skepticism toward prevention measures. Study findings may help tailor public health interventions and foster COVID-19 vaccination campaigns. Further studies considering the public primary sources of health information and health information seeking behaviors, as well as healthcare provider factors, are needed to further elucidate determinants of vaccination uptake.

## Figures and Tables

**Table 1 ijerph-19-13216-t001:** Date of the COVID-19 vaccine reservation opening for each age group in Tuscany.

Age Group	Date of the COVID-19 Vaccine Reservation Opening
over 80 years	1 January 2021
70–79 years	30 April 2021
60–69 years	10 May 2021
50–59 years	15 May 2021
40–49 years	21 May 2021
30–39 years	5 June 2021
over 16 years	12 June 2021

**Table 2 ijerph-19-13216-t002:** Sociodemographic factors and risk conditions of the whole sample and of the three vaccination subgroups (vaccinated on time, delayed vaccination, and unvaccinated).

Sociodemographic Factors and Risk Conditions	Whole Sample n (%)	Vaccinated on Time n (%)	Delayed Vaccination n (%)	Unvaccinated n (%)	*p*
**Total**	**691**	**510 (73.8%)**	**150 (21.7%)**	**31 (4.4%)**	
**Worker subgroups**					<0.001 *
Public Employees	209 (30.3%)	128 (61.2%)	66 (31.6%)	15 (7.2%)	
Civil Protection	482 (69.3%)	382 (79.3%)	84 (17.4%)	16 (3.3%)	
**Sex**					0.55 *
Female	270 (39.1%)	197 (73.0%)	63 (23.3%)	10 (3.7%)	
Male	421 (60.9%)	313 (74.3%)	87 (20.7%)	21 (5.0%)	
**Age**					0.02 °
18–39 years	196 (28.4%)	154 (78.6%)	33 (16.8%)	9 (4.6%)	
40–49 years	126 (18.2%)	86 (68.2%)	32 (25.4%)	8 (6.3%)	
50–59 years	166 (24.0%)	112 (67.5%)	50 (30.1%)	4 (2.4%)	
≥60 years	203 (29.4%)	158 (77.8%)	35 (17.2%)	10 (4.9%)	
**Education level**					0.60 *
Lower secondary school or less	280 (40.5%)	208 (74.3%)	58 (20.7%)	14 (5.0%)	
High school	288 (41.7%)	217 (75.3%)	61 (21.2%)	10 (3.5%)	
Bachelor Degree or higher	123 (17.8%)	85 (69.11%)	31 (25.20%)	7 (5.69%)	
**Nationality**					0.81 °
Italian	681 (98.6%)	503 (73.9%)	147 (21.6%)	31 (4.5%)	
Other	10 (1.4%)	7 (70.0%)	3 (30.0%)	0 (0%)	
**Smoking habits**					0.66 *
Current smokers	171 (24.7%)	125 (73.1%)	39 (22.8%)	7 (4.1%)	
Former smokers	141 (20.4%)	99 (70.2%)	33 (23.4%)	9 (6.4%)	
Never smokers	379 (54.9%)	286 (75.5%)	78 (20.6%)	15 (4.0%)	
**Risk conditions**					0.12 °
No risk condition	520 (75.2%)	379 (72.9%)	113 (21.6%)	28 (5.4%)	
One or more risk conditions	171 (24.8%)	131 (76.6%)	37 (28.2%)	3 (1.8%)	
**Influenza vaccination uptake in the 2019–2020 season**					<0.001 °
Yes	145 (21.0%)	126 (86.9%)	16 (11.0%)	3 (2.1%)	
No	546 (79.0%)	384 (70.3%)	134 (24.5%)	28 (5.1%)	
**Living with people aged 65 years or older or with people with chronic diseases**					1 *
Yes	198 (28.6%)	146 (73.7%)	43 (21.7%)	9 (4.5%)	
No	493 (71.4%)	364 (73.8%)	107 (21.7%)	22 (4.5%)	
**Number of cohabitants**					0.62 *
Living alone	64 (9.3%)	46 (71.9%)	13 (20.3%)	5 (7.8%)	
Living with no more than 2 people	412 (59.6%)	305 (74.0%)	88 (21.4%)	19 (4.6%)	
Living with more than 2 people	215 (31.1%)	159 (73.9%)	49 (22.8%)	7 (3.3%)	

* Chi-square test; ° Fisher’s exact test.

**Table 3 ijerph-19-13216-t003:** Health literacy level (inadequate HL: 1 ≤ x ≤ 2; problematic HL: 2 < x < 3; sufficient HL: 3 ≤ x ≤ 4) and scores of the whole sample and of the three vaccination subgroups: vaccinated on time, delayed vaccination, and unvaccinated.

Health Literacy	Whole Sample n (%)	Vaccinated on Time n (%)	Delayed Vaccination n (%)	Unvaccinated n (%)	*p*
Inadequate	46 (6.7%)	35 (6.9%)	9 (6.0%)	2 (6.5%)	0.61 *
Problematic	213 (30.8%)	151 (29.6%)	52 (34.7%)	10 (32.3%)	
Sufficient	337 (48.8%)	256 (50.2%)	71 (47.3%)	10 (32.3%)	
Mean	2.95 ± 0.59	2.96 ± 0.59	2.95 ± 0.58	2.87 ± 0.58	0.13 °
Median	3 (2.65–3.3)	3 (2.67–3.33)	3 (2.58–3.33)	2.75 (2.50–3.13)	

Missing value: 95 (13.7%) in the whole sample, 68 (13.3%) among Vaccinated on time, 18 (12%) among Vaccinated with a delay, 9 (29%) among Unvaccinated. * Fisher’s exact test; ° Kruskal–Wallis test.

**Table 4 ijerph-19-13216-t004:** Knowledge, attitudes, and practices towards COVID-19 preventive measures and COVID-19 risk perception of the whole sample and in the three vaccination subgroups (vaccinated on time, delayed vaccination, and unvaccinated).

KAPs and Risk Perception	Item	Vaccinated on Time		Delayed Vaccination		Unvaccinated		*p* *
		Mean ± SD	Median (IQR)	Mean ± SD	Median (IQR)	Mean ± SD	Median (IQR)	
**KNOWLEDGE**	Knowledge of prevention measures to protect from and avoid transmitting COVID-19	4.36 ± 0.77	5 (4–5)	4.21 ± 0.81	4 (4–5)	4.26 ± 0.93	5 (4–5)	0.10
**ATTITUDE**	Importance of the following measures to protect from and avoid transmitting COVID-19							
	washing hands on all recommended occasions	4.87 ± 0.42	5 (5–5)	4.82 ± 0.46	5 (5–5)	4.84 ± 0.45	5 (5–5)	0.47
	wearing a face mask	4.82 ± 0.50	5 (5–5)	4.71 ± 0.61	5 (5–5)	4.55 ± 0.89	5 (4–5)	0.006
	staying at home as much as possible	4.45 ± 0.94	5 (4–5)	4.51 ± 0.83	5 (4–5)	3.97 ± 1.20	4 (3–5)	0.03
	keeping a distance of at least one meter from other people	4.77 ± 0.59	5 (5–5)	4.75 ± 0.60	5 (5–5)	4.55 ± 0.72	5 (4–5)	0.07
	avoiding meeting friends/family members from another household, except for reasons of necessity	4.44 ± 0.90	5 (4–5)	4.45 ± 0.93	5 (4–5)	3.87 ± 1.18	4 (3–5)	0.003
	**Overall Attitude Score**	4.67 ± 0.50	5 (4.4–5)	4.65 ± 0.50	5 (4.4–5)	4.35 ± 0.70	4.6 (3.9–5)	0.01
**PRACTICES**	Adherence to the basic recommendations for COVID-19 prevention	4.56 ± 0.63	5 (4–5)	4.46 ± 0.66	5 (4–5)	4.39 ± 0.56	4 (4–5)	0.03
**RISK PERCEPTION**	How likely do you think you will contract COVID-19?	3.30 ± 1.11	3 (3–4)	3.28 ± 1.18	3 (3–4)	3.10 ± 1.33	3 (2–4)	0.80
	How dangerous do you think COVID-19 is for your health?	4.01 ± 1.07	4 (3–5)	3.87 ± 1.13	4 (3–5)	3.68 ± 1.28	4 (3–5)	0.20
	How afraid are you that your family members could contract COVID-19?	3.66 ± 1.06	4 (3–4)	3.55 ± 1.20	4 (3–4)	3.52 ± 1.43	4 (2–5)	0.80
	**Overall Risk Perception Score**	3.66 ± 0.80	3.67 (3.33–4.33)	3.57 ± 0.81	3.67 (3–4.25)	3.43 ± 1.14	4 (3–4.17)	0.40

* Kruskal–Wallis test. A multinomial logistic regression model was fitted including variables significantly associated with the outcome variables (*p* < 0.05) (Table 5). Since several attitude items were significantly associated with the outcome, the model was fitted including only the overall attitude score in order to avoid collinearity among variables.

**Table 5 ijerph-19-13216-t005:** Multinomial regression model: predictors of delayed vaccination and being unvaccinated for COVID-19.

Independent Variables (Predictors)	Delayed Vaccination *	*p*	Unvaccinated *	*p*
**Worker subgroups**				
Public Employees	Ref.		Ref.	
Civil Protection	0.51	0.002	0.33	0.009
**Age**				
18–39 years	Ref.		Ref.	
40–49 years	1.50	0.17	1.33	0.59
50–59 years	1.82	0.03	0.51	0.28
≥60 years	1.45	0.19	2.00	0.17
**Influenza vaccination uptake in the 2019–2020 season**				
Yes	Ref.		Ref.	
No	2.51	0.002	3.01	0.09
**KAP and Risk Perception (continuous)**				
Overall Attitude Score	1.06	0.76	0.47	0.01
Adherence to the basic recommendations for COVID-19 prevention	0.83	0.21	0.94	0.84

* Reference: Participants vaccinated on time.

## Data Availability

Data are available for scientific purposes after written request to the corresponding author.

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
