# Peer review of "Determinants of Actual COVID-19 Vaccine Uptake in a Cohort of Essential Workers: An Area-Based Longitudinal Study in the Province of Prato, Italy"

_ijerph, 2022, doi:10.3390/ijerph192013216_

Round 1
Reviewer 1 Report
In this study, the authors have aimed to explore determinants of COVID-19 vaccination in the Prato Province of Italy that could help in developing effective strategies for vaccine promotion. Overall, the study is well-performed and well-written. Although in this study, civil protection and public employee workers were included whose literacy rate is still higher and considering that the study is region-specific, if authors could include a few statements and/or data showing the vaccine hesitancy in unemployed or non-essential workers, that would also help readers to understand the difference in the perspective of the overall population.
Author Response
Reply: We thank the reviewer for this comments, we have added a paragraph discussing vaccine hesitancy in general population and in other population subgroups in order to better understand the difference in the perspective of the overall population (please see page 10, lines 283-290 of the revised manuscript)
Reviewer 2 Report
Interesting study. Appreciate if you can consider these few comments:
The abstract: results should have numerical values presented.
The introduction: there is much emphasis on HL while results did not fully support this. I think it is better to be more general incorporating other factors as well.
Methods: No data on reliability and validity of the tools. It is essential to add this accordingly.
Results: please avoid duplication between text and tables. Please provide reference and/or justification for HL categorisation presented in Table 3.
Discussion: it does not cover all the interesting findings. This section need to be extensively revised.
Overall, title used the term "uptake" and discussion used "hesitancy" and "refusal". I would appreciate to take a look on the terminology used as each one has its own meaning.
Thank you,
Author Response
- The abstract: results should have numerical values presented.
Reply: We thank the reviewer for this comment, we have added numerical values of the multinomial regression models presented in the abstract (please see the version of the abstract in the revised manuscript).
- The introduction: there is much emphasis on HL while results did not fully support this. I think it is better to be more general incorporating other factors as well.
Reply: We thank the reviewer for this comment, we have incorporated the HL in the paragraph where we describe all the other factors (please see page 2 lines 74-79 of the revised manuscript).
- Methods: No data on reliability and validity of the tools. It is essential to add this accordingly.
Reply: We thank the reviewer for this question. As for the HL measurement, the validated version of the HLS-EU-Q6 for the Italian language was used, we have better specified this in the method section (please see page 4 lines 150-152 and page 4 lines 177-179 of the revised manuscript). As for the “overall attitude score” and the “overall risk perception score”, these scores - although based on items frequently used in the literature - were not previously validated, we have acknowledged this limitation in the discussion section (please see page 11 lines 346-348). As far as the reliability and validity of data on COVID-19 vaccination, these data were retrieved from the official regional register that virtually covers all the COVID-19 vaccinations administered to the Tuscany Region residents. The only exception is the case in which COVID-19 vaccination occurred in other Italian regions or abroad; these vaccinations may have not been registered in case the participants had not transmitted the vaccination certificate to the regional authorities. This exception was already stated in the discussion section (please see page 11 lines 350-353 of the revised manuscript).
- Results: please avoid duplication between text and tables. Please provide reference and/or justification for HL categorization presented in Table 3.
Reply: We have revised the results according to the reviewer’s suggestion, furthermore we have provided the reference for HL categorization in the method section (please see page 4 lines 177-179 of the revised manuscript).
- Discussion: it does not cover all the interesting findings. This section need to be extensively revised.
Reply: We have further discussed our study findings, in particular we have added a paragraph concerning vaccination uptake in our cohort (please see page 10 lines 283-290).
- Overall, title used the term "uptake" and discussion used "hesitancy" and "refusal". I would appreciate to take a look on the terminology used as each one has its own meaning.
Reply: We thank the reviewer for this comment, we have aligned discussion’s terminology with the title.